**communications** engineering

# Smart polarization and spectroscopic holography for real-time microplastics identification
Yanmin Zhu[1], Yuxing Li [1], Jianqing Huang [1,2] & Edmund Y. Lam [1] ✉

Optical microscopy technologies as prominent imaging methods can offer rapid, non-destructive, non-invasive detection, quantification, and characterization of tiny particles. However, optical systems generally incorporate spectroscopy and chromatography for precise material determination, which are usually time-consuming and labor-intensive. Here, we design a polarization and spectroscopic holography to automatically analyze the molecular structure and composition, namely smart polarization and spectroscopic holography (SPLASH). This smart approach improves the evaluation performance by integrating multi-dimensional features, thereby enabling highly accurate and efficient identification. It simultaneously captures the polarization states-related, holographic, and texture features as spectroscopy, without the physical implementation of a spectroscopic system. By leveraging a Stokes polarization mask (SPM), SPLASH achieves simultaneous imaging of four polarization states. Its effectiveness has been demonstrated in the application of microplastics (MP) identification. With machine learning methods, such as ensemble subspace discriminant classifier, k-nearest neighbors classifier, and support vector machine, SPLASH depicts MPs with anisotropy, interference fringes, refractive index, and morphological characteristics and performs explicit discrimination with over 0.8 in value of area under the curve and less than 0.05 variance. This technique is a promising tool for addressing the increasing public concerning issues in MP pollution assessment, MP source identification, and long-term water pollution monitoring.

Microplastic (MP) particles, which may be ingested and remain in living organisms, have recently been reported to cause gastrointestinal dysmotility, obstruction, and death[1,2]. Plastics, as a significant source of water contamination[3], emitted 1.7 billion tonnes of greenhouse gases per year in 2015 and are expected to emit 6.5 billion tonnes by 2050, accounting for approximately 15% of the global carbon budget[4]. Optical microscopy technologies are the dominating methods for the detection and identification of MP, i.e., scanning electron microscope (SEM) and transmission electron microscope (TEM)[5,6], which capture the texture and morphological features of MPs. Nevertheless, the effectiveness of texture or morphological features in MPs' identification can be affected by various factors, such as environmental weathering and aging[7].

Accurate chemical determination by spectroscopies, such as Raman microscopy, and Fourier transform infrared spectroscopy (FT-IR), provide qualitative analysis for MP identification[8,9]. Molecular structural differences may result in the influence on absorption or reflection of the light source and

can be reflected in the spectroscopic information. Weak signals and the long processing time are several main drawbacks of spectroscopic methods[6,10]. Last but not least, sample preprocessing in some microscopic and spectroscopic methods is demanding and cumbersome, such as sample filtering, isolation, purification, dying[11,12], limiting their developments for in situ MPs' detection, early-stage MP pollution warning and real-time monitoring[7]. Therefore, promising methods, providing molecular structural and material-related features, are highly needed for MPs' identification.

Non-contact and non-invasive imaging systems, such as polarization imaging (PI) and digital holography (DH), can record physical features and may be alternative solutions for MPs identification. For example, PI captures the changes in the polarization state of the incident light introduced by the specimen. The polarization modification includes both phase and amplitude of the oscillating electric field vectors[13], related to the anisotropy and birefringence characteristics of the specimen[14,15]. However, the manual configuration of the polarization states in PI systems limits their usage in

[1]Department of Electrical and Electronic Engineering, The University of Hong Kong, Pokfulam, Hong Kong SAR, China. [2]Key Lab of Education Ministry for Power Machinery and Engineering, School of Mechanical Engineering, Shanghai Jiao Tong University, Shanghai, China. ✉e-mail: elam@eee.hku.hk

real-time field detection[16]. The polarization camera is an emerging technique that is specifically designed to detect the polarization state of incoming light. It typically comprises a conventional imaging sensor integrated with either a polarization filter array or a dedicated polarization sensor, for example, the division-of-focal-plane (DoFP) sensor[17]. In contrast to the Stokes mask-based polarization camera, the DoFP system incorporates a micro-lens array with the sensor. Leveraging advanced image reconstruction methods, the DoFP system achieves high imaging quality[18]. It has been applied in three dimensional HSI color space imaging and skylight polarization measurement[19,20]. It is worth noting that the application of polarization cameras for spectral function is still an area that requires further exploration.

DH is an emerging and advanced optical technology for small object detection[21–24]. It records the full complex wavefield with both amplitude and phase information[25] and is capable of measuring the morphological and optical parameters, such as the optical path difference (OPD) and refractive index (RI). In addition, DH is non-contact and non-invasive optical microscopy without the need for sample filtering and dyeing. Powered by machine learning (ML) and deep learning (DL)[26], DH presents outstanding capabilities in quick and accurate particle detection and analysis[27–29]. It was implemented as a portable device for in situ detection. However, single holographic features are easily influenced by environmental changes and weaken their reliability in MPs' identification[24,30,31]. Prior works with PI and DH classify MPs' categories based on the reconstructed holographic images and extract limited effective features[32,33]. Bianco et al.[24] proposed a method for identifying MPs in holographic images using fractal parameters. The primary and secondary fractal features were analyzed based on recorded data and sorted using ML methods. However, the experimental results revealed that the accuracy of identification was limited by morphological variations in the samples. The recorded features may not provide sufficient information for determining the material composition of MPs. Robust identification of MPs requires the inclusion of chemical or dominant structural-related features. Additionally, Bianco et al.[30] presented a related work in which they employed an off-axis DH system for MP identification. This method extracted a set of image features, including size, shape, and phase jumps. However, the identification performance of this method was compromised in real-world detection scenarios or when the holographic features were blurred. Valentino et al.[32] developed a polarization-sensitive holographic flow-cytometer for the determination of microfibers. In the polarization holographic system, interference is caused by two orthogonally polarized waves[34]. A polarization-sensitive material is required to record the polarization state of the light field, allowing for the precise retrieval of the object's amplitude and phase information. In this study, the effectiveness of the system is demonstrated in identifying fibers such as PA6, PA6.6, PET, PP, cotton, and wool. However, the system's capability has not been presented for *Daphnia magna*, *Chlorella*, stones, glasses, metal films, etc., which limits its practicality in real-world detections. Compared with the system in[32], our proposed system does not require an orthogonally polarized light source and is more compact, with fewer optical component requirements.

In this work, we firstly design a smart polarization and spectroscopic holography system, termed SPLASH, to simultaneously capture polarization, holographic, texture features, such as angle of polarization (AoP), degree of linear polarization (DoLP), phase retardation, and accomplish a molecular structure and composition-related discrimination without physically implementing a spectroscopic system. Rich features are extracted from the recorded synthetic experimental images and improves the system's discriminative power to distinguish among different MP materials, as well as other natural biological and microalgae specimens. In addition, SPLASH achieves four polarization states simultaneous imaging with a Stokes polarization mask (SPM) and automatic identification with ML methods. Qualitative and quantitative experimental results and analysis indicate the disclosed features as a physical fingerprint for material analysis. This work opens new routes for realizing spectroscopic holography and is a promising tool for addressing the increasing public concerning issues in assessing risks of MP pollution on ecosystems and human health, MP source identification, and long-term water pollution monitoring

## Results

### SPLASH visual inspection

Experimental images of microplastic materials with the polarization states of 0°, 45°, 90°, and 135° are presented in Fig. 1a–d for the imaging results visualization, which are recorded by the optical system in Fig. 1e. Selected MP specimens are polyethylene terephthalate (PET), polypropylene (PP), polycarbonate (PC), and polyvinyl chloride (PVC). The brightness in each polarization plot represents the intensity values in the underlying polarization states. The distinction of the intensity values and the holographic patterns in four polarization states performs the modulation in optical axis orientations and is related to the molecular structure and the optical properties of the material, such as birefringence, and anisotropy. Distinguishable characteristics of MP specimens demonstrate the discrimination capability of our system.

### Feature correlation analysis

Four categories of features are selected and grouped among the rich feature information encoded in SPLASH images. They are texture features, Fourier power spectrum features, holographic features, and polarization features, according to the feature description categories. Specifically, holographic features describe the holographic fringes' contrast and transparency. Polarization features describe the AoP, DoLP, phase retardation, and optical axis orientation values. Texture features include neighborhood gray-tone difference matrix[35] and gray level size zone matrix[36] features. Fourier power spectrum (FPS) features describe the radial summation and angular summation characteristics. Texture and FPS feature calculation definitions are presented in Supplementary Note S2 and S3.

We show the correlation circular plot of selected feature groups in Fig. 2a. The outside annular regions describe the feature categories. The thickness and color of the inside ribbons describe the correlation among feature groups. Thick and dark ribbons express high correlation values. Most polarization features, including optical axis orientation, phase retardation, and AoP present a positive correlation with texture, holographic, and FPS features. DoLP performs negative correlation values related to the part of texture features. One of the possible reasons is DoLP may fluctuate with the specimen thickness. Texture features vary in correlation values. Texture features are susceptible to weathering and aging process of the specimens, and are not defining characteristics in material identification. Transparency demonstrates a relatively strong negative correlation with other feature groups. This relation may be ascribed to that rich texture and structural information are prone to drop down the material transparency.

We present the violin plots of neighborhood gray tone difference matrix features, gray level size zone matrix features, and FPS features in Fig. 2b. Several texture features, for example, coarseness, strength, small zone emphasis, and zone size non-uniformity, show similar distributions among material specimen groups. Features, such as busyness, large zone high gray level emphasis, zone size entropy, perform specific feature distributions. After conducting data analysis and statistical evaluations, we summarize that sole discrimination with a single texture feature provides weak material identification. More physical or chemical features may improve the discrimination performance.

All the extracted features are set in calculating the correlation matrix. We inspect the correlation matrix in Fig. 2c. Detailed feature groups are as follows, C1: Neighborhood gray tone difference matrix features: Coarseness, Contrast, Busyness, Complexity, Strength; C2: FPS features: RadialSum, AngularSum; C3: Gray level size zone matrix features[36]; C4: Polarization features: Degree of linear polarization; C5: Polarization features: Angle of polarization; C6: Polarization features: Phase retardation; C7: Polarization features: Optical axis orientation; C8: Holographic features: Fringes contrast, Transparency. The violin plots of polarization as well as holographic features are presented in Fig. 2d. It is noticed that polarization features and holographic features demonstrate small variances compared with texture

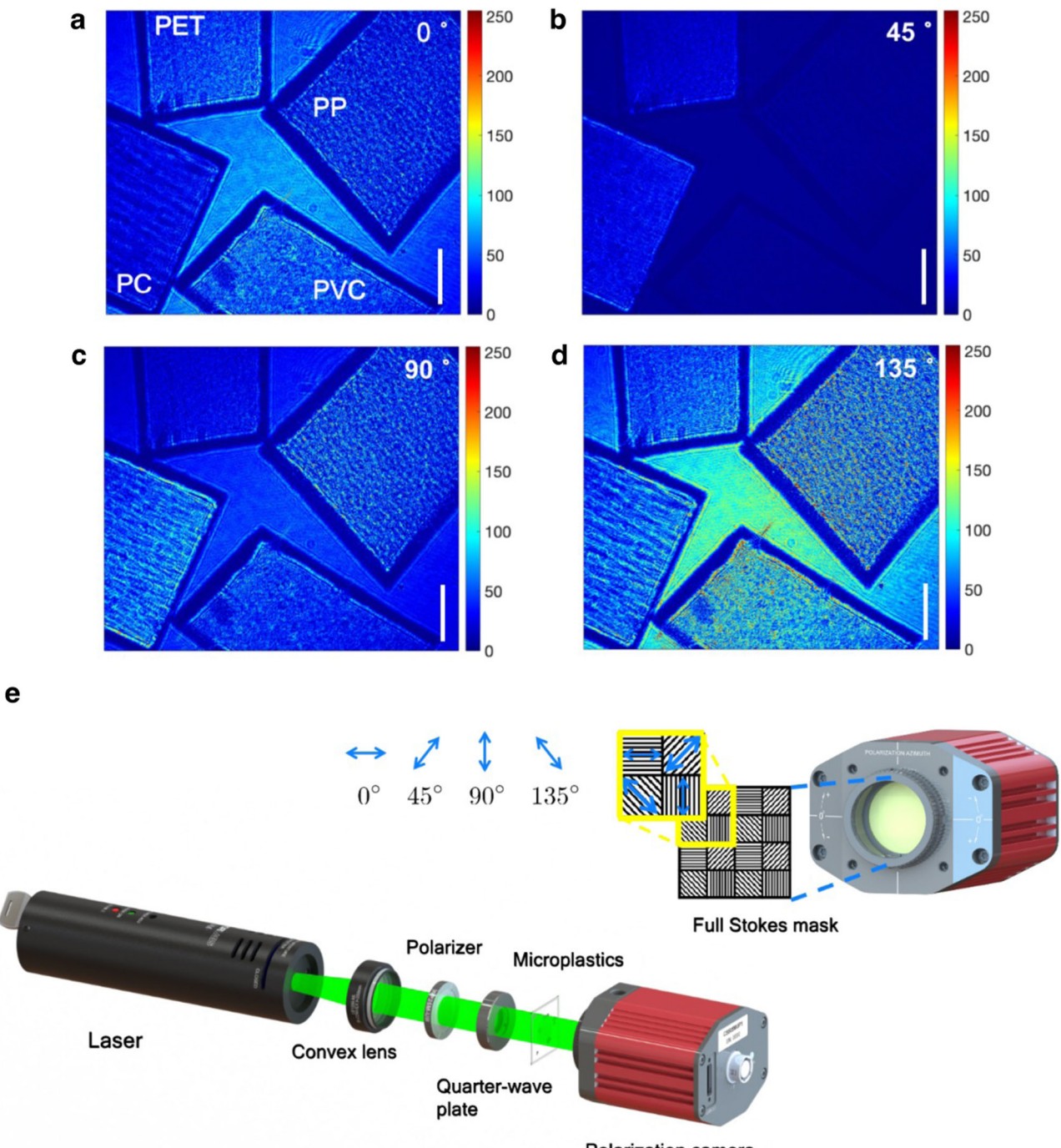

**Fig. 1 | Visualization for the experimental images with different material samples in.** (**a**) 0°, (**b**) 45°, (**c**) 90°, and (**d**) 135°. The scale bar is 0.5 mm. Colorbar is the value of the intensity. The material samples of polyethylene terephthalate, polypropylene, polycarbonate, and polyvinyl chloride perform various in the different polarization directions when passed through by the same incident light. (**e**) System schematic of SPLASH. Unpolarized light emitted by a laser diode with 532 nm wavelength. A convex lens is used for beam collimation. A linear polarizer is combined with a quarter-wave plate and inserted in the light path to adjust the laser intensity without introducing phase modulation. Then, the laser light goes through the sample plate, arriving at the sensing plane, and is analyzed by a polarization camera. A full Stokes polarization mask is mounted in front of the camera imaging sensing plane to record the sample images with four Stokes states ($S_0$, $S_1$, $S_2$ and $S_3$) in one shot.

features. Significant characteristics are presented in AoP, DoLP, and transparency.

## MP classification investigation
To inspect the method's capability for material analysis and MP identification, we train classifiers on the features of the MP dataset. Dataset consists 3221 images with a structure of 518 in PC class, 240 in PET class, 232 in PP class, 535 in PVC class, 573 in PMMA class, 208 in *Chlorella* class, 126 in *Daphnia magna* class, 789 in young root of plant T.S. class. Every image has a size of 1028 × 1232 pixels. Receiver operating characteristic (ROC) curves are plotted to evaluate the classification performance, quantitatively, as shown in Fig. 3a. Measurements of AUC and ROC are based on the results of 500 repeated training. All the classification tests used SVM classifiers. Random strategy separation and five-fold cross-validation are applied during the training stages[37]. Classification with polarization and weighted summation features gives higher TPR values than the classification with

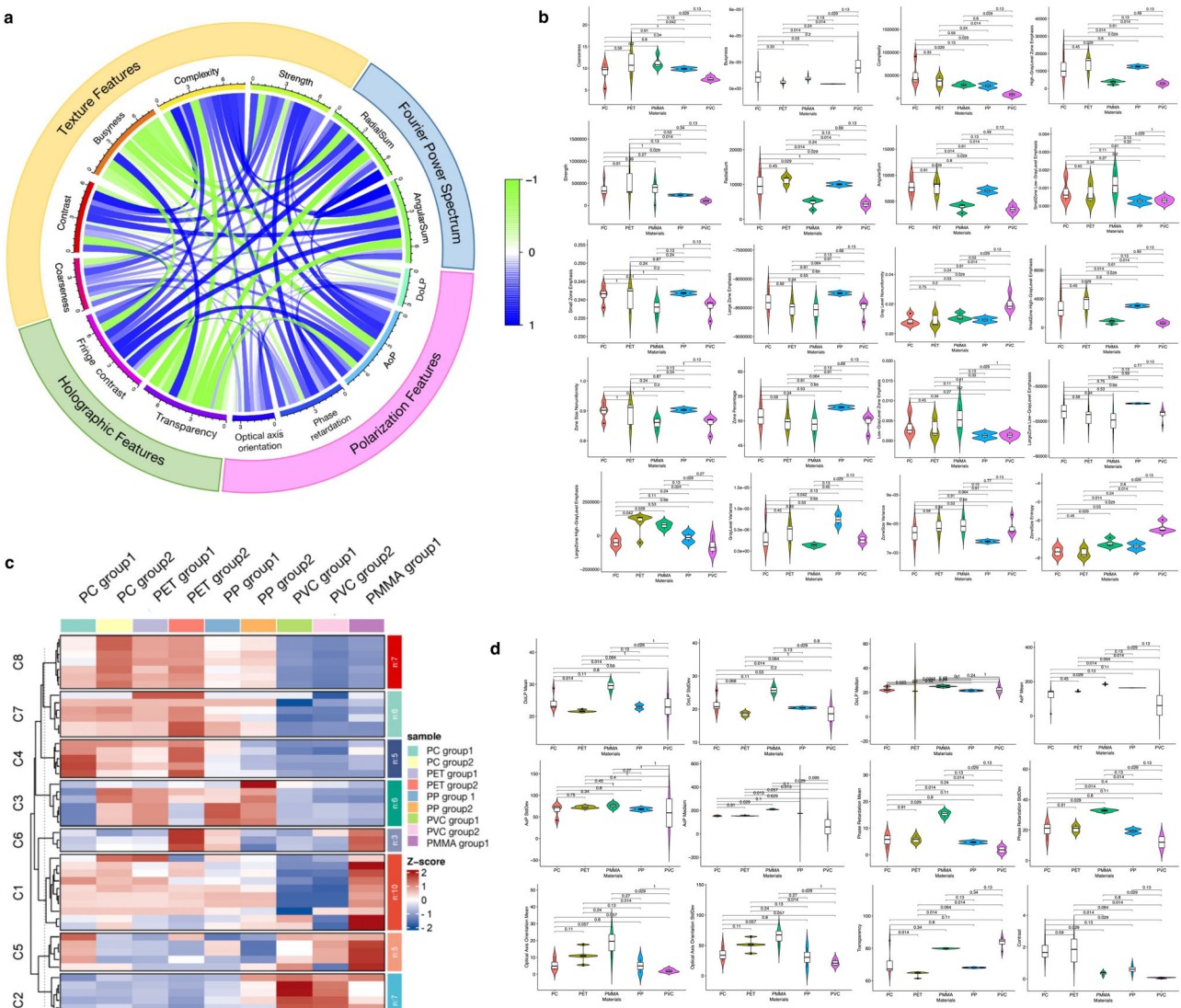

**Fig. 2 | Feature groups correlation and independence testing. a** Correlation circular plot for selected feature groups. Pearson correlation matrices are calculated among each of the feature groups. The thickness and color of the ribbons respond to the correlation values. Positive correlation shows in blue. The negative correlation shows in green. **b** Violin plots of different textures and Fourier power spectrum features show detailed differences between all five material considered categories. Inside box plots show the median and variance values. It is evident that texture features present various feature distributions for the same material categories. Parts of the features (i.e., coarseness, strength, small zone emphasis, zone size non-uniformity) show similar distribution among different material categories. **c** Correlation matrix of all selected features. Detailed feature groups are listed in the discussion section. **d** Violin plots of different polarization and holographic features show detailed differences between all five material considered categories. Material categories demonstrate less variance in polarization features compared with texture features.

texture and FPS features. A higher TPR value on a given FPR in the ROC plot demonstrates a better classifier capability. Another distinct note in the plot is that polarization, holographic, and weighted sum features have small variances and provide relatively reliable identification cues. The calculation equations for TPR and FPR are provided in Supplementary Note S5.

We further investigate the identification capability of features with different classifiers and present their area under the curve (AUC) values with a box plot in Fig. 3b. Four stable and general-used classifiers are chosen for experimental evaluation, which is ensemble subspace discriminant (ESD) classifier, k-nearest neighbors (KNN) classifier, neural network (NN), and support vector machine (SVM). AUC measures the whole two-dimensional area under the ROC curve. A higher AUC value evaluates a better classifier distinguishing capability[38]. In Fig. 3b, ESD classifier with texture features reaches the lowest accuracy slightly higher than 0.6. KNN, NN, and SVM perform higher AUCs which are around 0.65 and reach 0.7. The overall level of AUC values with texture features is in the range of 0.6 to 0.7. FPS features make a similar performance with a larger variance with the ESD classifier

and smaller variances with KNN, NN, and SVM classifiers. Classifications with holographic features have AUC values between 0.63 to 0.7. Classification with polarization features shows obvious improvements reaching 0.8. It is believed that polarization features offer aids in classification. The proposed SPLASH system and method provide discriminative features for MP identification and material analysis. Finally, we explore the classification performance on multi-dimensional features. Features are combined with a 4:2:2:2 ratio of polarization, holographic, texture, and FPS category, based on their single classification performance. It gives the best overall AUC performance at 0.85 with ESD, 0.83, 0.81, and 0.805 with KNN, NN, and SVM classifiers. The above results suggest that the polarization features are dominant in the classification tests.

**Discrimination experiments with natural particles**
Extension experiments with natural specimens are demonstrated to evaluate the distinguishing capability of the SPLASH system. In detail, specimens include the young root of plant T.S., *Chlorella*, and *Daphnia magna*. MPs are

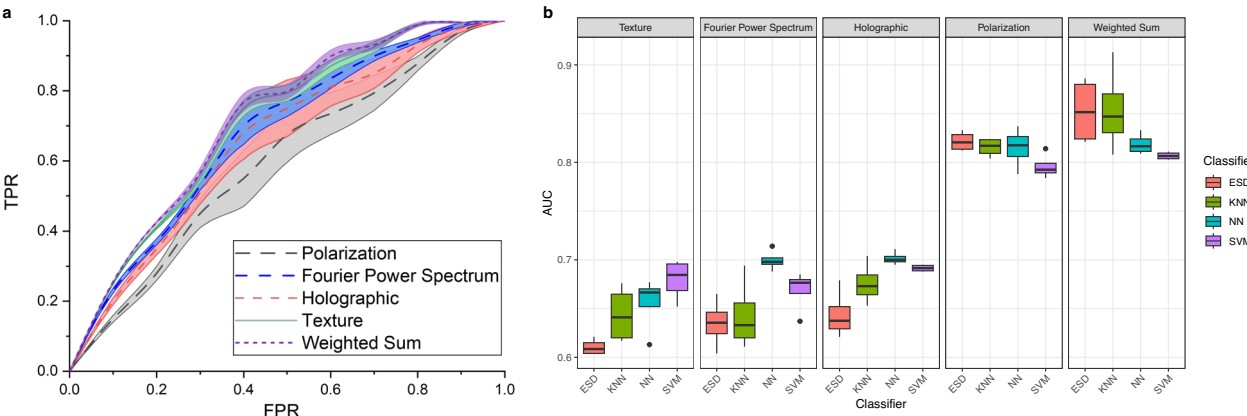

**Fig. 3 | Receiver operating characteristic (ROC) and classification area under the curve (AUC) results. a** ROC plot with extracted training features. Y-axis performs the true positive rate (TPR) and the x-axis gives the false positive rate (FPR). **b** Box plot for the classification AUC results with different classifiers, referring ensemble subspace discriminant (ESD) classifier, k-nearest neighbors (KNN) classifier, neural network (NN), and support vector machine (SVM).

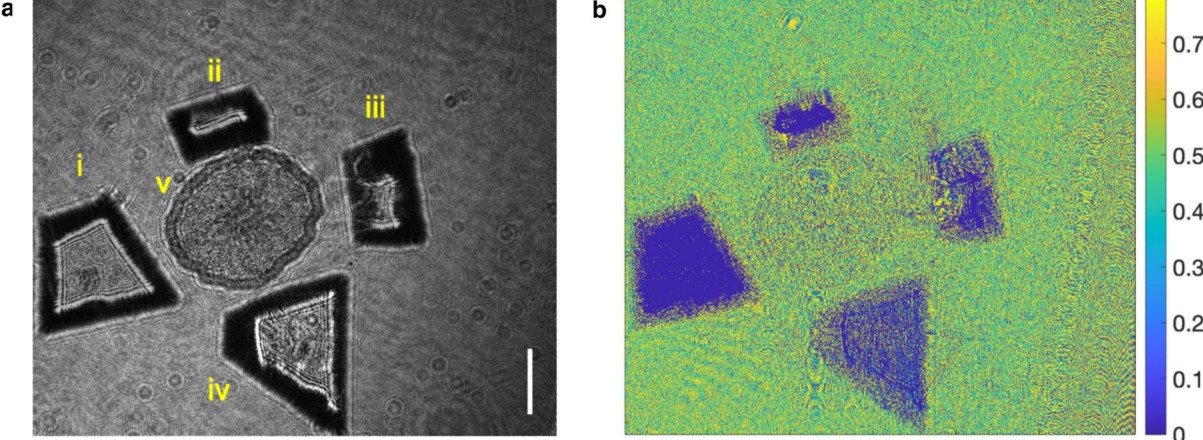

**Fig. 4 | Intensity image and angle of polarization plot of microplastics and young root of plant T.S. specimen. a** (i) polypropylene, (ii) polycarbonate, (iii) polyethylene terephthalat, (iv) polyvinyl chloride, and (v) young root of plant T.S. **b** Angle of polarization plot of the specimens in **a**. Scale bar: 0.5 mm.

found in aquatic and airborne environments[39]. They frequently mix with gravel, microorganisms, microalgae, and other natural particles[40,41]. Investigations on classifying MPs with natural materials and biological specimens can make contributions to the understanding of the distribution and abundance of microplastics in the environment, as well as their potential impacts on ecosystems and human health[42–44].

The experiments are conducted both in aquatic and airborne environments. We show the specimen intensity images and their AoP plots in Fig. 4 for the young root of plant T.S., Fig. 5 for *Chlorella* and Fig. 6 for *Daphnia magna*. In Fig. 4a, four kinds of MPs are presented, sequentially (i) PP, (ii) PC, (iii) PET and (iv) PVC. A young root of plant T.S. specimen (v) is placed in the middle. All the samples are placed on a colorless and transparent glass plate background. As we can see, the AoP plots, shown in Fig. 4b, demonstrate discriminate differences for qualitative material analysis. In Figs. 5 and 6, MP samples, *Chlorella* and *Daphnia magna* are in distilled water for image recording with alive *Chlorella* and *Daphnia magna*. In the field of view of Fig. 5a, specimens are (i) polystyrene (PS), (ii) *Chlorella*, and (iii) PET. The corresponding AoP plot is demonstrated in Fig. 5b for inspection. Figure 6 shows the discrimination among MPs and *Daphnia magna*. Sequentially, specimens are (i) PS, (ii) *Daphnia magna*, (iii) PP, and (iv) PC, as shown in Fig. 6a. Differences shown in AoP plots (Fig. 6b) among specimens are related to the distinctions in molecular structures between MPs and natural particles. MPs may provide a surface for microorganisms to attach to and grow on, potentially altering the microbial

community structure and function. Experiments are helpful for understanding the interactions between MPs and natural particles, and the potential impacts of microplastics on microbial communities and ecosystem processes[2].

## Discussion

For the first time, we have presented smart polarization and spectroscopic holography as a simultaneous polarization-texture-holographic characterization method for MPs' identification and discrimination with natural particles. This method combines molecular structure and composition-related image features, providing rich information for material analysis. The proposed optical system and method enable effective imaging analysis with spectroscopic capabilities, eliminating the need for a separate spectroscopic system. ML algorithms enhance system reliability and automation for large amounts of data processing. SPLASH is well-suitable for various environmental MPs' identification, making contributions to the understanding of MPs' migration, and interaction with micro-organisms in complex environments.

To evaluate the capability of SPLASH on MP identification, we inspect the experimental results of MP with different morphology characters and materials, combined with various natural particles, to simulate real-world MP existence situations. Specifically, classification experiments and tests are conducted among MPs, including PC, PET, PVC, and PP, with the young root of plant T.S., *Chlorella*, and *Daphnia magna*. To visualize the feature

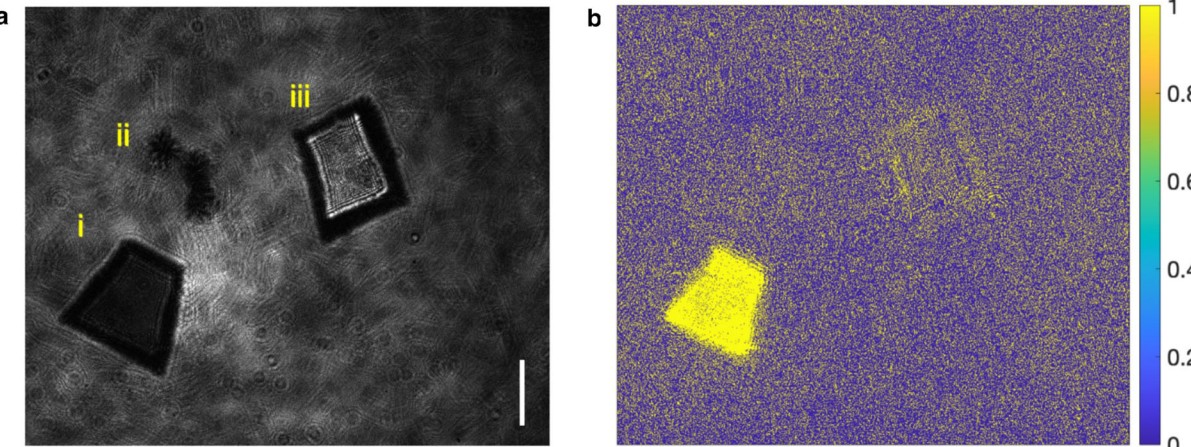

**Fig. 5 | Intensity image and angle of polarization plot of microplastics and *Chlorella*. (a)** (i) polystyrene, (ii) *Chlorella* and (iii) polyethylene terephthalate. **b** Angle of polarization plot of the samples. Scale bar: 0.5 mm.

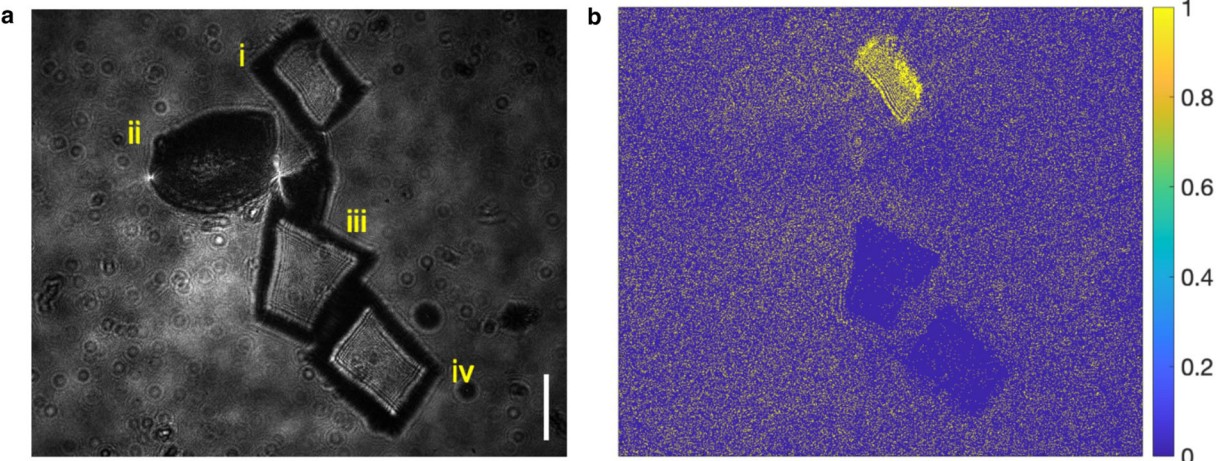

**Fig. 6 | Intensity image and angle of polarization plot of microplastics and *Daphnia magna*. a** (i) polystyrene, (ii) *Daphnia magna*, (iii) polypropylene and (iv) polycarbonate. **b** Angle of polarization plot of the samples. Scale bar: 0.5 mm.

distribution and correlation, we measure the correlation matrix among feature groups and present the violin plots of extracted MP features for PC, PET, PMMA, PP, and PVC. Classification tests show that polarization features and the weighted sum features provide over 0.8 AUC accuracies with less than 0.05 variance. Compared with the classification with texture and FPS features at 0.6 AUCs, polarization and holographic features enhance the system's discriminative powers and reliability for MPs' identification. Additional experiments with natural particles indicate that MPs can be identified by extracted features with plant specimens, microalgae, and micro-organisms.

Multi-dimensional evaluations and tests show a clear distinguishing capability among MPs and frequently encountered natural particles, making it a versatile technique for studying the distribution and impact of MPs in different environments. The non-destructive and non-invasive imaging setup provides an alternative for living and intact sample analysis. In addition, integrated ML methods are time and labor-saving[45]. The intelligent feature analysis and classification technique allows quick MPs identification, which is helpful for in situ MP detections. Multi-dimensional features, such as polarization, holographic, textural, morphological, and FPS, describe the molecular, structural, outline surface, and material-related characters of MPs to identify them with discriminative fingerprints. In the aquatic environment, image quality could be disturbed by the water scattering, especially in a turbid water environment. Image descattering

processing is a promising direction for future development[27]. By identifying the types and concentrations of MPs in different environments, researchers can obtain multi-dimensional assessments of the risk of MPs to human health and wildlife.

This system gives a reliable method for automatic MP identification. Future works could aim at developing high-throughput microfluidic[46] systems for non-contact, quick MP quantification[47,48]. Low-cost, compact, and portable devices are also of interest to researchers in the field and underwater MP detection[49]. Last but not least, the salient discriminative capability of SPLASH can be applied to industrial and medical material analysis, facilitating the identification of medicinal components and aiding in tissue diagnosis[50,51].

## Methods
### System setup
A compact optical system is designed in SPLASH, as shown in Fig. 1e. Unpolarized 532 nm laser light is emitted from a laser diode. A convex lens is placed in the light path for beam collimation. A linear polarizer is combined with a quarter-wave plate to adjust the light intensity without introducing phase modulation. The uniform-distributed circularly polarized light then goes through the sample plate and encodes the sample feature information with wavefront deformation. Formed patterns are finally recorded by a polarization camera (Crevis MG-A500P-22, monochrome, 2464 × 2056

resolution, $3.45\,\mu m \times 3.45\,\mu m$ pixel size). A PSM is mounted to capture images with full Stokes states (0˚, 45˚, 90˚, and 135˚) in a single shot. The spatial resolution of the recorded hologram is $1232 \times 1028$ pixels. The smallest detectable particle is $20\,\mu m$. The holographic polarization images with 4 polarization states are recorded simultaneously in a continuous recording format to realize the real-time data recording. The system takes around 2 ms to capture a group of data. This compact system optimizes the typical polarizer and analyzer pairs for modulation of the polarization states and eliminates manual adjustment of the polarizers during experiments. It improves the system imaging efficiency and offers a high-throughput hardware backbone. The omitting of lens pairs also reduces system aberration.

### Feature calculation, statistics analysis, and machine learning

Every experimental image recorded by the SPLASH system has been calculated and processed with feature extraction. As demonstrated in Section 2, multi-dimensional features are calculated at the aspect of texture, FPS, holographic, and polarization. We list the calculation equations of all feature categories in detail in Supplementary Note S1–S4.

The calculation and statistical analysis principles for ROC and AUC values are presented in Supplementary Note S5. The ensemble subspace discriminant (ESD) classifier, k-nearest neighbors (KNN) classifier, neural network (NN) classifier, and support vector machine classifier are trained with 5-fold cross-validation. MATLAB machine learning packages and classification learner toolbox are used for the assistance of training and predictions.

### MP and biological specimen preparation

PVC, PP, PS, PC, PET, and PMMA are purchased from Xinsheng Plastic Material Company, China. PVC, PP, PS, PC, PET, and PMMA specimens are in the size range of 1 to 3 mm. The young root of plant T.S. specimen is purchased from YuanHang Voyage, China, with the lab specimens set 100 PCs prepared microscope histology teaching slides. *Chlorella* contains *Chlorella salina*, *Chlorella pyrenoidosa*, and *Ankistrodesmus falcatus var.-tenuissimus Jao var. nov. Chlorella* and *Daphnia magna* are purchased from Benchongyiya Company, Hunan, China. *Daphnia magna* is cultured in a round transparent glass tank with a diameter of 20 cm, placed in a light-dark incubator with a light-to-dark ratio (*L*: *D*) of 16: 8. The light intensity is 3000 lx. The temperature is $(24 \pm 1)$ ˚C. *Daphnia magna* is fed by the *Chlorella* daily at 10 am and cultivated with tap water that has been aerated for more than 2 days. *Chlorella* is cultured in BG-11[52] medium. The culture condition is temperature $(25 \pm 1)$ ˚C. The light-to-dark ratio is 12h:12h. The light intensity is 5000-8000 lx. Shake the culture bottle 3–4 times a day, and switch every 1–2 weeks to make the algae grow into the logarithmic growth phase. During the whole cultivation process, all were sterilized. Including the natural particles and MP samples, this work presents a system detectable range in 0.6 mm–5 mm.

### Data availability

All data needed to replicate these results are available at https://github.com/ymzhu19eee/SPLASH.

### Code availability

All code needed to replicate these results is available at https://github.com/ymzhu19eee/SPLASH.

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

## Acknowledgements

The work is supported in part by the Research Grants Council of Hong Kong (GRF 17201620, RIF R7003-21). This work is partly funded by the Hong Kong Scholars Program (Grant No. XJ2022032). Jianqing Huang appreciates the partial financial support from Shanghai Jiao Tong University.

## Author contributions

Y. Z.: writing (original draft), writing (review and editing), sample preparation, experiments performing, formal analysis; Y. L.: writing (review and editing), sample preparation, experiments performing; J. H.: writing (review and editing), sample preparation, experiments performing; E. L.: writing (review and editing), resources, supervision, funding acquisition.

## Competing interests

The authors declare no competing interests
