## [Peer Review File · Communications Engineering]

Reviewers' comments:

Reviewer #1 (Remarks to the Author):

Comments

The manuscript "Smart polarization and spectroscopic holography (SPLASH) for real-time microplastics identification" concerns about the application in microplastics (MP) identification using the developed SPLASH system. The authors try to combine the polarization, holographic and texture features to distinguish among different MP materials, natural biological and microalgae specimens. The experimental results show that the MP identification can be well achieved by these features. In general, the experimental results of this study seem reliable.

However, the novelty and advantage of the presented SPLASH system in the manuscript are unclear. The optical design of the experimental setup is not novelty, and the use of the polarization camera is quite common in recent year. Moreover, the presented identification performances and the applications scenarios (such as the range of types and sizes that can be measured) are not quite better than other similar methods mentioned in the cited literature 26-30.

Point 1: From Fig.3 (b), the identification performance (AUC) seems that it mainly depends on the polarization features. The other features seem to be redundant or unimportant when using the machine learning algorithms. In other words, another potential better machine learning algorithm may be used to only train the polarization features, and the classification performance will be also comparable to or better than the current results.

Point 2: The "spectroscopic holography" in the title confuses me. If I understand correctly, some holographic features (Fringes contrast, Transparency) have been used in the identification, but it is not the real holography system like the cited literature 29. It seems not the combination of polarization and holography as I think.

Point 3: The mentioned (DoFP) polarization camera is quite common in recent year, such as Xiaobo Li, et al, "Fundamental precision limits of full Stokes polarimeters based on DoFP polarization cameras for an arbitrary number of acquisitions," Opt. Express 27, 31261-31272 (2019). In the manuscript, similar articles about the polarization camera are not mentioned and some of them should be cited to describe the experimental setup more clearly. Other references are shown below :

1. Skylight polarization measurement based on DoFP polarization camera.
2. An Angle of Polarization (AoP) Visualization Method for DoFP Polarization Image Sensors Based on Three Dimensional HSI Color Space.
3. Noise analysis in Stokes parameter reconstruction for division-of-focal-plane polarimeters.

Reviewer #2 (Remarks to the Author):

Authors are presenting a very promising technique for the detection and identification of microplastics

(MP) in water.

The main interesting points of the article are, in my opinion, the fact that the environment can be very complex, the number of features studied at once, the quality of the data analysis.

For these reasons, I am in favor of see this manuscript published. However, I have a few remarks:

- 1) What is the meaning of "symptoms" appearing at lines 58 and 89? Would it be characteristics of MPs?
- 2) Regarding the scale bar in all optical images, it seems that all the studied particle are larger than 5 mm (a few cm). This means that the particles are actually not microplastics, but macroplastics. This point is somewhat hidden in the manuscript and I would like to see a short sentence on the fact that the study is a proof of concept that could lead to the identification of MPs.
- 3) What is the actual resolution of the imaging system and what would be the smallest detectable particle?
- 4) The background seems to have an important role in the quality of the picture. Could this be discussed?
- 5) How fast is the analysis of the data? Since the device is intended to measure in real time, this is important to discuss this point.
- 6) I suggest to carefully review the writings, complexity of sentences, and English. Just to make clearer and more impactful the innovation of the study.

November 18, 2023

Communications Engineering Editorial Office

Dear Editor,

Authors' reply to the reviewers' comments

("Smart polarization and spectroscopic holography (SPLASH) for real-time microplastics identification", manuscript ID: COMMS-ENG-23-0302)

We would like to thank the constructive comments from the reviewers. We have made corresponding revisions in the manuscript. In the following, our response and a list of revisions in the revised manuscript will be provided (The original comments of the reviewers are written in italic).

Comments of Reviewer 1:

The manuscript "Smart polarization and spectroscopic holography (SPLASH) for real-time microplastics identification" concerns about the application in microplastics (MP) identification using the developed SPLASH system. The authors try to combine the polarization, holographic and texture features to distinguish among different MP materials, natural biological and microalgae specimens. The experimental results show that the MP identification can be well achieved by these features. In general, the experimental results of this study seem reliable.

Answer: We appreciate your acknowledgement of the experimental results and the methodology employed in our work.

However, the novelty and advantage of the presented SPLASH system in the manuscript are unclear. The optical design of the experimental setup is not novelty, and the use of the polarization camera is quite common in recent year. Moreover, the presented identification performances and the applications scenarios (such as the range of types and sizes that can be measured) are not quite better than other similar methods mentioned in the cited literature 26-30.

Answer: We appreciate your thorough reading of the manuscript. The polarization cameras as an emerging technology are increasingly useful with the development of advanced optical systems. These cameras are capable of capturing polarization images under specific polarization states, which are helpful for analyzing the polarization characteristics of detected samples. However, to the best of our knowledge, polarization cameras are only used in polarization holography. There is no “polarization imaging + digital holography” multimodal imaging system with a polarization camera to achieve real-time imaging. The polarization holography and the polarization-sensitive holography have the capability to record and retrieve the amplitude, phase and polarization of light in a polarization sensitive recording material. In polarization holography, two orthogonally polarized waves are used as the light source, forming an interference field with the periodic change of the polarization states [1]. A polarization-sensitive material is required for the imaging recording, such as the photo-anisotropic material.

[1] Kakichashvili, S.D. Method for phase polarization recording of holograms. *Sov. J. Quantum Electron.* 1974, 4, 795.

Our system is the first published “polarization + digital holography” system based on a polarization camera. In our system, there is no requirement for the orthogonally polarized light source, or a photo-anisotropic material imager. Our proposed method enables analysis of material composition, morphology, and structural characteristics and achieves spectroscopic-like functionality for composition determination without the need for a physically implemented spectroscopic system. Specifically, our method offers several advancements compared to [26]-[30]. **The bibliography is updated in the revised manuscript. We paste the original bibliography for your reference.**

26 . Bianco, V., Pirone, D., Memmolo, P., Merola, F. & Ferraro, P. Identification of microplastics based on the fractal properties of their holographic fingerprint. *ACS Photonics* 8, 2148–2157, (2021).

27 . Bianco, V. et al. Microplastic identification via holographic imaging and machine learning. *Adv. Intell. Syst.* 2, 1900153 (2020).

28 . Zeng, T., Zhu, Y. & Lam, E. Y. Deep learning for digital holography: A review. *Opt. Express* 29, 40572–40593 (2021).

29 . Valentino, M. et al. Intelligent polarization-sensitive holographic flow-

cytometer: Towards specificity in classifying natural and microplastic fibers. *Sci. The Total. Environ.* **815**, 152708 (2022).

30 . Behal, J. et al. Toward an all-optical fingerprint of synthetic and natural microplastic fibers by polarization-sensitive holographic microscopy. *ACS Photonics* **9**, 694–705 (2022).

- In literature [26], Bianco, V., Pirone, D., Memmolo, P., Merola, F., and Ferraro, P. proposed a method for identifying microplastics based on the fractal properties of their holographic fingerprint in their article titled 'Identification of microplastics based on the fractal properties of their holographic fingerprint' published in *ACS Photonics* **8**, 2148–2157 (2021). The authors utilized an off-axis digital holographic setup for their optical system, which did not integrate any polarization imaging component. They analyzed the primary and secondary fractal features based on recorded data and sorted them using machine learning techniques. However, the experimental results revealed that the identification accuracy was limited by the morphological variations of the samples. The recorded features may not provide sufficient information for determining the material composition of microplastics. Robust identification of microplastics requires the inclusion of chemical or dominant structural-related features.
- Literature [27], Bianco, V. et al. Microplastic identification via holographic imaging and machine learning. *Adv. Intell. Syst.* **2**, 1900153 (2020). [27] is a former and related work with [26]. An off-axis digital holographic system is designed for microplastic identification. Compared with the method in [26], this method extracts fewer categories of image features, including sizes, shapes, and phase jumps. The identification performance is eliminated in the real-world detection scenes or when the holographic features are blurred.
- In literature [28], deep learning-based digital holography methods are introduced. This review presents the leading-edge methods and technologies that combine deep learning to preprocess or post-process digital holographic images. Various learning strategies are discussed, including transfer learning, zero-shot learning, and physical-prior model-based networks. Additionally, the review includes detailed diagrams of digital holographic systems, encompassing both traditional setups and specifically

designed configurations. However, the polarization holographic system, particularly with a polarization camera, is not included or reviewed by [28]. The current manuscript introduces a material analysis and microplastic identification method that pioneers the use of a multi-modal imaging system, leveraging polarization, refractive index, morphology, and holographic features for multi-dimensional sample material determination

- Literature [29] presents a polarization-sensitive holographic flow-cytometer used for microfiber determination. A pair of orthogonal polarized light is set as the light source for the image recording. Experimental results demonstrate its effectiveness in identifying PA6, PA6.6, PET, PP, cotton, and wool fibers. However, the polarization state control is required for the light source. Furthermore, the system's capability has not been confirmed or demonstrated for microorganisms, micro-algae, stones, glass, metal film, etc. These natural particles are frequently encountered during the identification of microplastics in real-world experiments or field detections. Our work extensively demonstrates the system's capability in real-world detection scenarios.
- In literature [30], the authors constructed an off-axis holographic system with a polarizer and detector to collect both sample polarization and holographic characteristics. A linear horizontal and a linear vertical laser light are set as the light source. Compared to our proposed system, this setup requires accurate polarization state control, more optical components, and is less compact. Additionally, our detection samples encompass a wider range of categories, including *Daphnia magna*, *Chlorella*, glass, metal films, and more. These sample categories consist of natural particles and represent real-world samples, making our experiments highly relevant for field detection. In our paper, we calculate morphological, holographic, and polarization state characteristics to identify microplastics with different feature categories. In contrast, [30] focuses on calculating polarization characters, their eigenvectors, and eigenvalues for microplastics identification. Our method provides a more comprehensive set of dimensional features.

We add the comparisons with [26]-[30] in Section 1. The manuscript is revised as follows:

DH is an emerging and advanced optical technology for small object detection [21]-[24].

It records the full complex wavefield with both amplitude and phase information [25] and is capable of measuring the morphological and optical parameters, such as the optical path difference (OPD) and refractive index (RI). In addition, DH is non-contact and non-invasive optical microscopy without the need for sample filtering and dyeing. Powered by machine learning (ML) and deep learning (DL) [26], DH presents outstanding capabilities in quick and accurate particle detection and analysis [27]-[29]. It was implemented as a portable device for *in situ* detection. However, single holographic features are easily influenced by environmental changes and weaken their reliability in MPs' identification [30]-[32]. Prior works with PI and DH classify MPs' categories based on the reconstructed holographic images and extract limited effective features [33,34]. Bianco *et al.* [24] proposed a method for identifying MPs in holographic images using fractal parameters. The primary and secondary fractal features were analyzed based on recorded data and sorted using ML methods. However, the experimental results revealed that the accuracy of identification was limited by morphological variations in the samples. The recorded features may not provide sufficient information for determining the material composition of MPs. Robust identification of MPs requires the inclusion of chemical or dominant structural-related features. Additionally, Bianco *et al.* [31] presented a related work in which they employed an off-axis DH system for MP identification. This method extracted a set of image features, including size, shape, and phase jumps. However, the identification performance of this method was compromised in real-world detection scenarios or when the holographic features were blurred. Valentino *et al.* [33] developed a polarization-sensitive holographic flow-cytometer for the determination of microfibers. In the polarization holographic system, interference is caused by two orthogonally polarized waves [35]. A polarization-sensitive material is required to record the polarization state of the light field, allowing for the precise retrieval of the object's amplitude and phase information. In this study, the effectiveness of the system is demonstrated in identifying fibers such as PA6, PA6.6, PET, PP, cotton, and wool. However, the system's capability has not been presented for *Daphnia magna*, *Chlorella*, stones, glasses, metal films, etc., which limits its practicality in real-world detections. Compared with the system in [33], our proposed system does not require an orthogonally polarized light source and is more compact, with fewer optical component requirements.

Point 1: From Fig.3 (b), the identification performance (AUC) seems that it mainly

depends on the polarization features. The other features seem to be redundant or unimportant when using the machine learning algorithms. In other words, another potential better machine learning algorithm may be used to only train the polarization features, and the classification performance will be also comparable to or better than the current results.

Answer: Thank you for your constructive suggestion. The classification results only trained on the polarization features can be found in Figure 3(b) the fourth column. The results range from 0.78 to 0.82. The classification results with all features range from 0.81 to 0.85. The polarization features are dominant in the classifications. However, morphological, holographic, and polarization multi-modal features provide robust and higher AUC results. We add the classification results based on polarization features by using more kinds of machine learning classifiers in the following table for your reference. With various and better machine learning algorithms, the weighted sum features present higher AUCs than the performances of polarization features. We add the comparisons and analysis in Section 2.3.

Method	medium tree	coarse tree	quadratic discriminator	Gaussian naive Bayes	kernel naive Bayes
polarization features	0.78	0.77	0.77	0.81	0.81
weighted sum features	0.82	0.83	0.82	0.86	0.84

(1)

Point 2: The “spectroscopic holography” in the title confuses me. If I understand correctly, some holographic features (Fringes contrast, Transparency) have been used in the identification, but it is not the real holography system like the cited literature 29. It seems not the combination of polarization and holography as I think.

Answer: We appreciate your comprehensive reading and reviewing. The holographic system in literature [29] is an off-axis holographic system. The reference beam and object beam go along different light paths before reaching the sample imaging plane. The holographic system in our polarization holographic setup is based on the frame of in-line digital holography and it is a more compact holographic system. In in-line holographic

system, the object light and reference light go along the same light path before reaching the object imaging plane. With the reference of Katz, Joseph, and Jian Sheng. "Applications of holography in fluid mechanics and particle dynamics." *Annual Review of Fluid Mechanics* 42 (2010): 531-555., Kim, Myung K. "Principles and techniques of digital holographic microscopy." *SPIE reviews* 1.1 (2010): 018005., and Goodman, Joseph W. *Introduction to Fourier optics*. Roberts and Company publishers, 2005., as described in [28], in-line and off-axis systems are all typical digital holographic setups. The phase and amplitude information are recorded by the spatial coherence and reconstructed by the computational imaging processing. In our system, as shown in Figure 1(e), a coherent light source, i.e., a laser, goes through a convex lens to expand the light beam diameter. A polarizer and a quarter-wave plate are combined and balance the light intensity of every polarization direction. The polarization camera has a CMOS imaging sensor and a full-Stokes mask. In the holographic system part, the laser light goes along the same light path for the object and reference light. The holographic patterns, encoding the object information, are recorded by the CMOS imaging sensor in the polarization camera. The proposed system is based on an in-line digital holographic system and polarization imaging. Compared with the off-axis DH system, which is shown in [29], the in-line DH system has a more compact setup and benefits for in-the-field and underwater systems.

28 Zeng, T., Zhu, Y. & Lam, E. Y. Deep learning for digital holography: A review. *Opt. Express* 29, 40572–40593 (2021).

29 Valentino, M. et al. Intelligent polarization-sensitive holographic flow-cytometer: Towards specificity in classifying natural and microplastic fibers. *Sci. The Total. Environ.* 815, 152708 (2022).

Point 3: The mentioned (DoFP) polarization camera is quite common in recent year, such as Xiaobo Li, et al, "Fundamental precision limits of full Stokes polarimeters based on DoFP polarization cameras for an arbitrary number of acquisitions," Opt. Express 27, 31261-31272 (2019). In the manuscript, similar articles about the polarization camera are not mentioned and some of them should be cited to describe the experimental setup more clearly. Other references are shown below: 1. Skylight polarization measurement based on DoFP polarization camera. 2. An Angle of Polarization (AoP) Visualization Method for DoFP Polarization Image Sensors Based

on Three Dimensional HSI Color Space. 3. Noise analysis in Stokes parameter reconstruction for division-of-focal-plane polarimeters.

Answer: We appreciate your comprehensive reading and constructive suggestions. We added the citations of the above-mentioned literature in bibliography [17]-[20] and provided the comparisons in the revised manuscript Section 1. The manuscript is revised as follows:

Non-contact and non-invasive imaging systems, such as polarization imaging (PI) and digital holography (DH), can record physical features and may be alternative solutions for MPs identification. For example, PI captures the changes in the polarization state of the incident light introduced by the specimen. The polarization modification includes both phase and amplitude of the oscillating electric field vectors [13], related to the anisotropy and birefringence characteristics of the specimen [14,15]. However, the manual configuration of the polarization states in PI systems limits their usage in real-time field detection [16]. **The polarization camera is an emerging technique that is specifically designed to detect the polarization state of incoming light. It typically comprises a conventional imaging sensor integrated with either a polarization filter array or a dedicated polarization sensor, for example, the division-of-focal-plane (DoFP) sensor [17]. In contrast to the Stokes mask-based polarization camera, the DoFP system incorporates a micro-lens array with the sensor. Leveraging advanced image reconstruction methods, the DoFP system achieves high imaging quality [18]. It has been applied in three dimensional HSI color space imaging and skylight polarization measurement [19,20]. It is worth noting that the application of polarization cameras for spectral function is still an area that requires further exploration.**

13 Goldstein, D. H. Polarized Light (CRC press, 2017)

14 Song, S., Kim, J., Hur, S., Song, J. & Joo, C. Large-area, high-resolution birefringence imaging with polarization-sensitive fourier ptychographic microscopy. ACS Photonics 8, 158–165 (2021).

15 Tsvetkov, V. N. Reviews of topical problems: Flow birefringence and the structure of macromolecules. Sov. Phys. Uspekhi 6, 639–681 (1964).

16 Graydon, O. Imaging polarization. Nat. Photonics 343 (2013).

17 Li, X., Hu, H., Goudail, F. & Liu, T. Fundamental precision limits of full Stokes polarimeters based on DoFP polarization cameras for an arbitrary number of acquisitions. Opt. Express 27, 31261–31272 (2019).

- 18 Bai, C., Jiang, Z., Zhao, J., Wu, S. & Zhang, Q. Noise analysis in Stokes parameter reconstruction for division-of-focal-plane polarimeters. *Appl. Opt.* 61, 7084–7094 (2022).
- 19 Wang, H. et al. An angle of polarization (AoP) visualization method for DoFP polarization image sensors based on three dimensional HSI color space. *Sensors* 19, 1713 (2019).
- 20 Li, S., Xu, W., Zhan, J., Xu, G. & Song, G. Skylight polarization measurement based on DoFP polarization camera. In *Eighth Symposium on Novel Photoelectronic Detection Technology and Applications*, vol. 12169, 499–504 (SPIE, 2022)

Comments of Reviewer 2:

Authors are presenting a very promising technique for the detection and identification of microplastics (MP) in water.

The main interesting points of the article are, in my opinion, the fact that the environment can be very complex, the number of features studied at once, the quality of the data analysis.

For these reasons, I am in favor of see this manuscript published. However, I have a few remarks.

Answer: Thanks for your detailed reading and appreciation of our work. We've made clarifications and revisions according to your helpful suggestions.

1) What is the meaning of "symptoms" appearing at lines 58 and 89? Would it be characteristics of MPs?

Answer: Thanks for the insightful comments. Symptoms are the characteristics of samples. We apologize for the unclear illustration in the original manuscript. We revised the illustrations as follows: For line 58 in original manuscript now change to: Distinguishable characteristics of MP specimens demonstrate the discrimination capability of our system.

Line 89 in original manuscript now change to: Significant characteristics are presented in AoP, DoLP, and transparency.

The manuscript is revised accordingly.

2) Regarding the scale bar in all optical images, it seems that all the studied particle are larger than 5 mm (a few cm). This means that the particles are actually not microplastics, but macroplastics. This point is somewhat hidden in the manuscript and I would like to see a short sentence on the fact that the study is a proof of concept that could lead to the identification of MPs.

Answer:

Thank you for your constructive suggestion. The samples in the result images demonstrated in the original manuscript are in the range of 0.6 mm to 5 mm. We add the description of the sample size in Section 4.3 to show this work can lead to the identification of MPs. We corrected the typo in the caption of Figure 1.

Section 4.3 is revised as follows:

PVC, PP, PS, PC, PET, and PMMA are purchased from Xinsheng Plastic Material Company, China. **PVC, PP, PS, PC, PET and PMMA specimens are in the size range of 1 to 3 mm.** The young root of plant T.S. specimen is purchased from YuanHang Voyage, China, with the lab specimens set 100 PCs prepared microscope histology teaching slides. *Chlorella* contains *Chlorella salina*, *Chlorella pyrenoidosa*, and *Ankistrodesmus falcatus var.tenuissimus Jao var. nov.* *Chlorella* and *Daphnia magna* are purchased from Benchongyiya Company, Hunan, China. *Daphnia magna* is cultured in a round transparent glass tank with a diameter of 20 cm, placed in a light-dark incubator with a light-to-dark ratio ($L : D$) of 16 : 8. The light intensity is 3000 lx. The temperature is $(24 \pm 1)^\circ\text{C}$. *Daphnia magna* is fed by the *Chlorella* daily at 10 am and cultivated with tap water that has been aerated for more than 2 days. *Chlorella* is cultured in BG-11 medium. The culture condition is temperature $(25 \pm 1)^\circ\text{C}$. The light-to-dark ratio is 12h:12h. The light intensity is 5000-8000 lx. Shake the culture bottle 3-4 times a day, and switch every 1-2 weeks to make the algae grow into the logarithmic growth phase. During the whole cultivation process, all were sterilized. **Including the natural particles and MP samples, this work presents a system detectable range in 0.6 mm - 5 mm.**

3) *What is the actual resolution of the imaging system and what would be the smallest detectable particle?*

Answer: Thank you for your comments. The spatial resolution of the imaging sensor is 2464×2056 with a square pixel size of $3.45 \mu\text{m}$. The spatial resolution of the recorded hologram is 1232×1028 pixels. The smallest detectable particle is $20 \mu\text{m}$. We add the system resolution and the smallest detectable particle size description in Sec. 4.1.

Section 4.1 is revised as follows:

A compact optical system is designed in SPLASH, as shown in Fig.1(e). Unpolarized 532 nm laser light is emitted from a laser diode. A convex lens is placed in the light path for beam collimation. A linear polarizer is combined with a quarter-wave plate to adjust the light intensity without introducing phase modulation. The uniform-distributed circularly polarized light then

goes through the sample plate and encodes the sample feature information with wavefront deformation. Formed patterns are finally recorded by a polarization camera (Crevis MG-A500P-22, monochrome, 2464×2056 resolution, $3.45 \mu\text{m} \times 3.45 \mu\text{m}$ pixel size). A PSM is mounted to capture images with full Stokes states ($0^\circ, 45^\circ, 90^\circ$ and 135°) in a single shot. **The spatial resolution of the recorded hologram is 1232×1028 pixels. The smallest detectable particle is $20 \mu\text{m}$. The holographic polarization images with 4 polarization states are recorded simultaneously in a continuous recording format to realize the real-time data recording. The system takes around 2 ms to capture a group of data.** This compact system optimizes the typical polarizer and analyzer pairs for modulation of the polarization states and eliminates manual adjustment of the polarizers during experiments. It improves the system imaging efficiency and offers a high-throughput hardware backbone. The omitting of lens pairs also reduces system aberration.

4) *The background seems to have an important role in the quality of the picture. Could this be discussed?*

Answer: Thank you for your suggestions. We record the images in air and water environments. In the water, the images may be degraded with the water scattering effects. The images are more blurred than in the air. As shown in Figure 4, 5 and 6, Figure 4 is recorded in the air and on a glass plate. Figure 5 and 6 are recorded in the water environment. Our system demonstrates the capability to identify the MPs both in the air and water environment. We add the background discussions in Section 2.4 and Section 3.

Section 2.4 is revised as follows:

Extension experiments with natural specimens are demonstrated to evaluate the distinguishing capability of the SPLASH system. In detail, specimens include the young root of plant T.S., *Chlorella*, and *Daphnia magna*. **MPs are found in aquatic and airborne environments.** They frequently mix with gravel, microorganisms, microalgae, and other natural particles. Investigations on classifying MPs with natural materials and biological specimens can make contributions to the understanding of the distribution and abundance of microplastics in the environment, as well as their potential impacts on ecosystems and human health.

The experiments are conducted both in aquatic and airborne environments. We show the specimen intensity images and their AoP plots in Figure 4 for the young root of plant T.S., Figure 5 for *Chlorella* and Figure 6 for *Daphnia magna*. In Figure 4(a), four kinds of MPs are presented, sequentially (i) PP, (ii) PC, (iii) PET and (iv) PVC. A young root of plant T.S. specimen (v) is placed in the middle. **All the samples are placed on a colorless and transparent glass plate**

background. As we can see, the AoP plots, shown in Figure 4(b), demonstrate discriminate differences for qualitative material analysis. In Figure 5 and Figure 6, MP samples, *Chlorella* and *Daphnia magna* are in the distilled water for image recording with alive *Chlorella* and *Daphnia magna*. In the field of view of Figure 5, specimens are (i) PS, (ii) *Chlorella*, and (iii) PET. The corresponding AoP plot is demonstrated in Figure 5(b) for inspection. Figure 6 shows the discrimination among MPs and *Daphnia magna*. Sequentially, specimens are (i) PS, (ii) *Daphnia magna*, (iii) PP, and (iv) PC. Differences shown in AoP plots among specimens are related to the distinctions in molecular structures between MPs and natural particles. MPs may provide a surface for microorganisms to attach to and grow on, potentially altering the microbial community structure and function. Experiments are helpful for understanding the interactions between MPs and natural particles, and the potential impacts of microplastics on microbial communities and ecosystem processes.

5) How fast is the analysis of the data? Since the device is intended to measure in real time, this is important to discuss this point.

Answer: We appreciate your comprehensive reading and constructive suggestions. The recording time for a group of polarization holographic images is <2 ms. The training time for the machine learning methods is in the range of 1 min to 5 min. With the trained functions, MPs' identification are achieved in less than 5s. We add the discussion of the system speed in Section 4.1.

6) I suggest to carefully review the writings, complexity of sentences, and English. Just to make clearer and more impactful the innovation of the study.

Answer: We appreciate your suggestions. We have carefully reviewed the writings and polished the sentences of the manuscript to make clearer and more impactful the innovation of the study.

We are grateful to the editor and the reviewers for their great work and valuable suggestions. We hope the reply is useful to illustrate what we have changed in our revision according to the comments from the editor and the reviewers.

Yours sincerely,

Yanmin Zhu, Yuxing Li, Jianqing Huang, and Edmund Y. Lam

REVIEWERS' COMMENTS:

Reviewer #1 (Remarks to the Author):

The manuscript has been well revised.

Reviewer #2 (Remarks to the Author):

The manuscript has been well revised, and I do not see any other comments to be done.

December 18, 2023

Communications Engineering Editorial Office

Dear Editor,

Authors' reply to the reviewers' comments

("Smart polarization and spectroscopic holography for real-time microplastics identification", manuscript ID: COMMSENG-23-0302A)

We would like to thank the constructive comments from the reviewers. We make the correspond response for the reviewer's comments. (The original comments of the reviewers are written in italic).

Comments of Reviewer 1:

The manuscript has been well revised.

Answer: We appreciate your careful reviews and helpful suggestions for the manuscript revision.

Comments of Reviewer 2:

The manuscript has been well revised, and I do not see any other comments to be done.

Answer: We appreciate your careful reviews and helpful suggestions for the manuscript revision.

Yours sincerely,

Yanmin Zhu, Yuxing Li, Jianqing Huang, and Edmund Y. Lam

Department of Electrical and Electronic Engineering, University of Hong Kong

School of Mechanical Engineering, Shanghai Jiao Tong University